# Reversing Dysdynamism to Interrupt Mitochondrial Degeneration in Amyotrophic Lateral Sclerosis

**DOI:** 10.3390/cells12081188

**Published:** 2023-04-19

**Authors:** Gerald W. Dorn

**Affiliations:** Department of Internal Medicine (Pharmacogenomics), Washington University School of Medicine, St. Louis, MO 63110, USA; gdorn@wustl.edu

**Keywords:** mitochondrial fusion, mitochondrial fission, mitochondrial transport, mitophagy

## Abstract

Amyotrophic lateral sclerosis is one of several chronic neurodegenerative conditions in which mitochondrial abnormalities are posited to contribute to disease progression. Therapeutic options targeting mitochondria include enhancing metabolism, suppressing reactive oxygen production and disrupting mitochondria-mediated programmed cell death pathways. Herein is reviewed mechanistic evidence supporting a meaningful pathophysiological role for the constellation of abnormal mitochondrial fusion, fission and transport, collectively designated mitochondrial dysdynamism, in ALS. Following this is a discussion on preclinical studies in ALS mice that seemingly validate the idea that normalizing mitochondrial dynamism can delay ALS by interrupting a vicious cycle of mitochondrial degeneration, leading to neuronal die-back and death. Finally, the relative benefits of suppressing mitochondrial fusion vs. enhancing mitochondrial fusion in ALS are speculated upon, and the paper concludes with the prediction that the two approaches could be additive or synergistic, although a side-by-side comparative trial may be challenging to perform.

## 1. Introduction

Amyotrophic lateral sclerosis (ALS) is the most common motor neuron-specific neuropathy. Both late onset (~6th–7th decade) sporadic ALS and earlier onset familial ALS are characterized by progressive degeneration of upper and lower motor neurons. Clinically, the disease presents with fasciculations and slurred speech, leading gradually to dysphagia and paralysis. Death typically occurs within 5 years of symptom onset, frequently from complications of aspiration pneumonia and respiratory failure [1,2,3].

Many potential causes have been identified for ALS, but the underlying mechanistic etiology remains poorly defined. Sporadic ALS, accounting for approximately 90% of all cases, may be an autoimmune disorder and has been correlated with such diverse environmental factors as neurotoxin ingestion in Guam and military service in the Middle East [4,5]. Familial ALS, which represents only ~10% of total ALS, has been linked to mutations of over a dozen different genes, most commonly *superoxide dismutase 1* (*SOD1*), *TAR DNA binding protein 43* (*TDP-43*), *fused in sarcoma* (*FUS*) and *C9orf72* [6]; mutations of these genes can also be detected in subjects who present with sporadic ALS. Clinically, ALS overlaps with frontotemporal dementia (FTD), which is characterized by degeneration of the frontal and temporal lobe cortices. Indeed, the *C9orf72* G4C2 (GGGGCC) repeat expansion mutation has been identified in ~40% of familial ALS and 25% of familial FTD [7,8]

Taken together, accumulated genetic and clinical data point to a combination of defective neuronal free oxygen radical management, proteostasis, axonal transport and/or mitochondrial transport and fitness as causal or contributory factors in ALS [9]. This paper examines the therapeutic potential of correcting mitochondrial dysdynamism, i.e., the reduced fusion, exaggerated fission and/or impaired motility, that are observed in ALS. Explored herein is the concept that interventions targeting dysdynamism can generally improve mitochondrial fitness and resistance to—or recovery from—injury, and can therefore interrupt a vicious cycle of mitochondrial degeneration and diminish neuronal die-back in ALS, independent of the principal underlying etiology.

## 2. Neuronal Mitochondria—A Special Case

The primary function of mitochondria in normal cells is to employ oxidative phosphorylation of ADP via the electron transport chain to produce ATP. Compared to non-excitable cells such as fibroblasts (a standard experimental platform for studying mitochondrial dynamics [10,11,12]), neurons are especially dependent upon mitochondrial ATP because of the high energy requirements for electrochemical neurotransmission [13]. In ALS, the requirement for constant ATP production may be compounded by altered or impaired mitochondrial metabolism, as reported in pre-clinical models [14,15,16]. The unique elongated anatomy of upper and lower motor neurons poses special challenges for ATP delivery, which may contribute to motor neuron selectivity of ALS. Because ATP undergoes spontaneous hydrolysis in aqueous solution at physiological pH, mitochondria need to be positioned at subcellular locations proximate to ATP utilization [17]. Accordingly, mitochondrial residence within the distal synaptic branches of normal neurons (where ATP production fuels neuronal signaling) and in the terminal growth buds of damaged/regenerating axons (where ATP production fuels neuronal repair and regrowth) is essential for neuronal homeostasis. Since the vast majority (>99%; approximately 1100) of mitochondrial proteins are encoded by the cell nucleus, translated in proximate ribosomes and then imported into mitochondria [18], the generation of “new” mitochondria to replace those lost to damage or senescence necessarily occurs in neuronal soma. Thus, the mitochondrial supply chain is long and fragile. Using lower motor neurons as an example, the production of mitochondria via replicative fission and biogenic maturation (vide infra) takes place in the ventral horns of the spinal cord, but mitochondrial ATP production is required in neuromuscular synapses and all along connecting axons that may be up to a meter in length. Directed mitochondrial transport from neuronal soma through axons and to the periphery is therefore essential to fuel neuronal transmission via local ATP generation. 

Like everything, mitochondria wear out. Because >99% of their protein components are transcribed in the nucleus and translated in neuronal soma, the functional lifespan of a mitochondrion located at a distal synaptic terminus can be abbreviated [19]. Moreover, as neurons are terminally differentiated cells, mitochondrial renewal during cell proliferation does not occur [20,21]. Mitochondrial senescence is manifested in part by damage to respiratory enzymes and the uncoupling of oxidative phosphorylation from ATP synthesis, resulting in the elaboration of ROS [22]. For this reason, the formidable physical challenges facing the anterograde transport of young, healthy mitochondria from the spinal cord to distal motor neuron synapses are balanced by the equal challenge of removing damaged, potentially cytotoxic mitochondria from distal neuronal termini through retrograde transport. Indeed, accumulating evidence suggests that the targeted elimination of neuronal mitochondria via mitophagy requires their mobilization from neuronal termini back to the soma [23], or even their externalization from neurons to neighboring cells for degradation [24]. The process by which mitochondria are transported anterograde and retrograde along microtubules through neuronal processes involves interactions of mitochondrial outer membrane Miro proteins with cytosolic TRACK adaptor proteins that tether individual mitochondria to kinesin and dynein motor proteins (reviewed in [25,26,27]). Given the unique dependence of long spinal and peripheral nerves upon mitochondrial transport, it is not surprising that mitochondrial dysmotility and clumping within neuronal soma (like traffic congestion, a sign of impaired transport) are observed in many neurodegenerative diseases, including ALS [28,29,30,31]. Decreased Miro1 expression, as reported in ALS patient spinal cords, may directly contribute to mitochondrial dysmotility in this disease [32,33].

Effective neuronal mitochondrial transport that refreshes the distal mitochondrial workforce by removing impaired mitochondria and delivering fresh, fully functional replacements may have special importance in ALS neurons because key mitochondrial metabolic pathways can be impaired in this condition [34,35]. Fortunately, biological redundancy has provided for mechanisms of mitochondrial repair and renewal that are less taxing to the cell than whole organelle removal and replacement. Chief among alternate mitochondrial repair mechanisms is organelle fusion, in which a non-critically impaired mitochondrion fuses and exchanges contents, structural components and mtDNA with one or more healthy neighbors [36]. As a consequence of organelle fusion, damaged components within the impaired mitochondrion are diluted, while potentially damaging mtDNA variants are corrected through complementation with donated normal mtDNA. In contrast to fusion-mediated mitochondrial repair that appears to be a ubiquitous mechanism for mitochondrial renewal, impaired mitochondrial motility preferentially impacts neurons possessing long cellular processes requiring vigorous mitochondrial transport to and from the periphery. Nevertheless, both defective mitochondrial transport and an imbalance between reparative mitochondrial fusion and opposing mitochondrial fission are hallmarks of degenerative neurological conditions, including ALS [37,38,39]. 

Taken together, the central importance of mitochondria for neuronal signaling and repair, the dual effects of mitochondria as the source of ATP that drives essential biological processes and of ROS that damages cellular components, and the prevalence of mitochondrial abnormalities in etiologically diverse neurological diseases, suggest that mitochondrial pathology can significantly contribute to neurodegeneration as a secondary consequence of multiple different primary pathophysiological lesions. An attractive hypothesis that may explain why mitochondrial involvement in neurodegeneration is so prevalent considers that the primary underlying causal factor, be it a misfolded or aggregated protein, increased ROS or some other cellular stress, damages the neuronal mitochondrial collective. The possible role played by ROS in ALS has received a great deal of attention, and the challenges and opportunities of therapeutically targeting ROS have recently been comprehensively reviewed [40]. Related to the current topic of mitochondrial dysdynamism, it is posited that damaged individual mitochondria can accumulate to such a degree that normal compensatory mechanisms, such as reparative mitochondrial fusion and mitochondrial removal by mitophagy, are overwhelmed. Indeed, both of these processes are reportedly impaired in ALS and other neurodegenerative conditions [37,41]. This situation can initiate a “mitochondrial contagion” [42], wherein a subpopulation of damaged mitochondria collaterally injures their healthy counterparts by elaborating ROS that oxidatively damages respiratory enzymes, organelle membranes, mtDNA and mRNA [43,44,45,46]. This creates a feed-forward cycle of mitochondrial degeneration in ALS [47] that can either trigger a mitochondrial release of cytochrome c to activate caspase-mediated apoptosis [48], or evoke mitochondrial depolarization sufficient for transitioning of the mitochondrial permeability pore to induce non-apoptotic mitochondria-mediated programmed cell death [49,50]. In either scenario, primary mitochondrial damage evokes secondary mitochondrial damage that causes neuronal drop-out (Figure 1). While there are abundant data indicating that the above individual events occur in ALS, it remains unclear whether mitochondrial abnormalities in non-mitochondrial neurological disorders are simply a consequence of the primary underlying neuronal pathology that contribute little to disease progression, or if they can become a substantial contributory factor. Quantifying the impact of secondary and tertiary mitochondrial injury on disease progression cannot be satisfactorily addressed in studies of cultured cells or isolated organelles. Rather, it requires mitochondria-targeted therapeutic interventions performed using in vivo animal models of the subject human diseases, as reviewed below.

## 3. Mitochondrial Dysdynamics as a Therapeutic Focus in Neurodegeneration

As introduced above, mitochondrial dysmotility and fragmentation are widely observed in ALS and other neurodegenerative conditions. The pervasiveness of mitochondrial phenotypes in etiologically diverse neurological conditions suggests convergent pathophysiological mechanisms, with an opportunity for broadly applicable therapeutic interventions. Indeed, a meta-analysis of mitochondria-targeted interventions in clinical and animal models of ALS found that the conceptual approach of improving mitochondrial dysfunction prolonged survival in (largely SOD1 mutant) ALS [51]. Attempts to address mitochondrial components of ALS have included mitigating the effects of increased ROS [52,53,54] and suppressing apoptosis [39,55,56]. Because these approaches have had difficulty translating to substantial disease modification in the clinic [40,57,58], it is worth considering alternative lines of attack. The most frequently reported mitochondrial abnormality in ALS is organelle shortening or “fragmentation”, i.e., a decrease in the relationship between mitochondrial length and width (most commonly reported as mitochondrial “aspect ratio”). It is intuitively obvious that mitochondrial shortening can be produced by an altered balance between mitochondrial fusion that produces longer organelles vs. mitochondrial fission that generates shorter and rounder daughters. However, the presence of small, round neuronal mitochondria in static images of ALS neurons does not provide insight into the underlying abnormality, i.e., whether the aberrancy is diminished fusion or increased fission or a combination [59]. In ALS, decreased expression of fusion-promoting mitofusins (MFN) and increased expression of the fission mediator dynamin-related protein 1 (DRP1/DMNL1) seem to suggest that the disease evokes a general imbalance in mitochondrial dynamics [60], perhaps contributed to by increased ROS [61,62]. Further support for this notion is provided by frequent observations of impaired mitochondrial motility in ALS, perhaps attributable to the diminished expression of mitochondrial outer membrane Miro and MFN proteins [32,33]. Notwithstanding a lack of clarity on the precise molecular abnormalities responsible for mitochondrial dysdynamism, the concept of repairing abnormal mitochondrial morphology by restoring the proper balance between mitochondrial fusion and fission, and of restoring normal mitochondrial motility by any means, was attractive and testable using animal models of ALS [38,63,64]. The results of several particularly impactful preclinical studies are discussed below in the context of our current understanding of the mechanisms leading to, and the pathophysiological role played by, mitochondrial dysdynamism.

### 3.1. Targeting Mitochondrial Fission

Mitochondrial fission, the division of a single parent mitochondrion into two daughter organelles, is the process by which normal mitochondria replicate and abnormal mitochondria rid themselves of damaged components. Replicative fission is symmetric, producing equally sized and normally functioning daughter organelles that mature and grow both by fusing with other normal mitochondria (or, in cell types having interconnected mitochondria, with the network) and by incorporating newly generated nuclear- and mtDNA-encoded constituent proteins. The combination of replicative fission and the incorporation of newly synthesized biological components is termed mitochondrial biogenesis [65]. In contrast, asymmetric mitochondrial fission of non-lethally damaged mitochondria is a reparative process that directs the most severely damaged components into the smaller of two different-sized daughter organelles. The concentration and sequestration of dysfunctional elements within one daughter mitochondrion causes it to depolarize, triggering its selective mitophagic elimination in part via the PINK-Parkin pathway [66]. The remaining daughter mitochondrion, into which the healthy components were directed, re-joins the mitochondrial collective through fusion or biogenesis, as described for the healthy daughters of a replicative fusion event (Figure 2). Thus, mitochondrial fission is essential both for mitochondrial quantity control via replication and mitochondrial quality control via mitophagy. The major factors and events that mediate mitochondrial fission have been identified and are described in detail elsewhere [47,67]. Briefly, cytosolic DRP1 is recruited to mitochondria where it oligomerizes in a chain of pearls structure that circumscribes the mitochondrion and contracts, essentially garroting the organelle in two [68]. Cytosolic DRP1 may bind to mitochondrial Fis1 [69,70]. However, there appear to be Fis1-independent mechanisms for mitochondrial fission, i.e., other mitochondrial DRP1 receptors such as mitochondrial fission factor (MFF) [71,72]. The Fis1-DRP1 interaction has been posited to have specific relevance to pathological or stress-induced mitochondrial fission [73], whereas Mff-DRP1 interactions may mediate physiological/replicative fission [72]. Alternately, Fis1 may contribute to mitochondrial fragmentation by suppressing MFN and OPA1-mediated mitochondrial fusion [74].

The evidence that mitochondrial fragmentation in the neurons of ALS patients and other neurodegenerative diseases is caused, at least in part, by pathologically increased mitochondrial fission is compelling [60,75,76,77,78,79,80]. Importantly, suppressing mitochondrial fission with the small molecule DPR1 inhibitor Mdivi-1 [81], or with the P110 peptide that competitively inhibits the DRP1-Fis1 interaction [82], has reversed in vitro mitochondrial fragmentation provoked by ALS-linked mutations of SOD1, TDP43 or FUS1 [60,83]. These results prompted proof-of-concept studies evaluating the idea that suppressing mitochondrial fission in ALS would reverse mitochondrial fragmentation and/or delay neuromuscular degeneration in vivo. Indeed, Mdivi-1 administered intra-peritoneally to a small group (*n* = 3) of mice expressing SOD1 G93A in foot muscle reversed mitochondrial dysmotility and improved mitochondrial fragmentation in those muscles [60]. However, it is important to note that Mdivi-1 lacks specificity and may have DRP1-independent cytoprotective side effects [84,85]. Thus, the role of DRP1-mediated mitochondrial fission is not unambiguously established using this reagent.

The P110 peptide inhibitor of Drp1-Fis1 interactions was employed in a detailed and comprehensive in vivo evaluation of ALS modification by mitochondrial fission inhibition [83]. The peptide was administered by a subcutaneously implanted osmotic mini pump to SOD1 G93A mice from the onset of the neurodegenerative phenotype, at approximately 90 days of age. Functional endpoints were measured after 10 and 28 days of treatment; some mice were followed until they achieved a pre-terminal state to assess survival. P110-treated ALS mice exhibited improved mobility and grip strength compared to their age-matched vehicle-treated counterparts. The ventral horns of P110-treated ALS mice (wherein motor neuron soma are located) showed evidence of mitochondrial retention, and ALS mouse distal hindlimb muscles exhibited myocyte preservation with less severe mitochondrial structural abnormalities, compared to vehicle-treated mice. Finally, survival to the pre-terminal state was increased by 10 days (from 122 to 132 days, or ~8%) in P110-treated mice. Taken together, the P110 and Mdivi studies support the notion that inhibiting mitochondrial fission can delay phenotype progression in, and modestly prolong the survival of, SOD1 G93A mice. Indeed, P110 has also moderated disease in a murine model of HD [86], suggesting possible general applicability for suppressing mitochondrial fission in genetic neurodegeneration exhibiting mitochondrial dysdynamism. These pioneering studies stimulated an interest in testing other clinically translatable means of normalizing mitochondrial dysdynamism in ALS and related conditions. 

### 3.2. Targeting Mitochondrial Fusion

Mitochondrial fusion, the physical integration of an individual mitochondrion with one or more neighbors, is considered to be a major mechanism for mitochondrial repair by complementation and dilution [36] (Figure 2). Mitochondrial fusion is initiated when MFN1 and MFN2 proteins located on outer mitochondrial membranes adjust their conformation to extend across cytosolic space and dimerize in trans with MFN proteins on adjacent mitochondria. The result is a reversible joining of the two mitochondria; conceptually, this is much like Velcro. The macromolecular organization of MFN proteins in cis on a given mitochondria, and in trans between mitochondria, is unclear, as are the precise protein structures of “inactive” and “active” MFN. However, evidence is accumulating that phosphorylation can be a major factor determining MFN protein conformation and ability to promote mitochondrial fusion (and, by inference, the antecedent physical association or “tethering” of two mitochondria) [36,87,88]. Although MFN1 and MFN2 are dynamin family GTPases, mitochondrial tethering mediated by their association in trans is GTP-independent. However, subsequent outer membrane fusion, mediated also by MFN1 and/or MFN2, requires MFN GTPase activity and is irreversible. For this reason, naturally occurring and artificial MFN mutants lacking catalytic GTPase activity evoke organelle association/tethering without fusion, which can be visualized as mitochondrial “clumping”. In human patients, several GTPase-defective and other loss-of-function MFN2 mutants cause the sensory-motor peripheral neuropathy, Charcot–Marie–Tooth disease type 2A (CMT2A) [89,90,91]. Because mitochondria are compartmentalized by an inner mitochondrial membrane, outer membrane fusion must be followed by inner membrane fusion mediated by the related dynamin family GTPase, optic atrophy 1 (OPA1). Loss-of-function OPA1 mutants cause another neurological condition, dominant optic atrophy (DOA) [92]. Intriguingly, CMT2A and DOA sometimes overlap [91,93,94,95], suggesting that the consequences of a primary impairment in mitochondrial fusion from whatever genetic cause preferentially damage peripheral and retinal neurons. There are varied reports of decreased MFN and OPA1 expression in ALS [78,96,97,98]. Accordingly, Wang et al. employed a genetic approach to enhance neuronal mitofusin activity in the murine SOD1 G93A mouse [97]. Based on their observation that Mfn2 expression was diminished specifically in the lumbar spinal cord of SOD1 G93A mice, and that spinal cord-targeted ablation of Mfn2 provoked neuronal die-back that resembled ALS, SOD1 G93A mice were bred to mice transgenically expressing Mfn2 in neurons. Life-long Mfn2 overexpression (i.e., from before birth and therefore prior to disease) delayed ALS onset and prolonged the survival of SOD1 G93A mice. A pathophysiological role proposed for Mfn2-mediated calpastatin transport on mitochondria is somewhat controversial [87] and would be difficult to distinguish from the general improvement in mitochondrial motility, transport and delivery to distal neuronal termini provoked by mitofusin activity [87,99,100,101,102,103,104,105]. Nevertheless, these studies confirmed previous findings that neuronal Mfn2 ablation adversely impacts neuronal function in vivo [106,107,108], while demonstrating that enhanced mitochondrial fusion can delay phenotype progression in this model.

Genetic manipulation to mitigate disease in animal models, as described above, has been useful for experimental proof of principle but is difficult to translate to the clinic. Transgenic overexpression or gene ablation in the germ line are currently not feasible in human subjects. While viral vectors may ultimately be used to express cDNAs (to correct loss of function) or inhibitory RNAs (to correct gain of function) in human disease, their clinical utility remains limited. Thus, medical interventions for chronic clinical conditions overwhelmingly favor the use of small molecule pharmaceuticals. It is therefore fortunate that multiple small molecules have recently been described which may favorably impact the disequilibrium between mitochondrial fusion and fission in neurodegeneration [87,99,100,102,103,109]. 

Recent feasibility studies demonstrating the benefits of pharmacological mitofusin activation in ALS have employed three chemically distinct but functionally similar small molecule mitofusin activators that activate both MFN1 and MFN2 through a common allosteric mechanism [87,102,103]. The Chimera class of mitofusin activators are the original small molecule mitofusin activators. These compounds have a triazol/urea-containing backbone giving rise to functional groups that mimic the hydrophobic and hydrogen donor moieties of amino acid side-chains critical to the peptide–peptide interaction that controls mitofusin conformation [87]. Thus, Chimera class mitofusin activators bind to MFN1 and MFN2 and disrupt endogenous peptide–peptide interactions that maintain a folded/closed mitofusin conformation, thereby increasing the probability of the open conformation favoring fusion and motility. The prototype, Chimera C, has been useful for determining if mitochondrial abnormalities manifested in a variety of neurodegenerative conditions can be reversed through mitofusin activation. To date, Chimera C has improved mitochondrial fragmentation in metabolically stressed dermal fibroblasts from patients with hereditary ALS [105], CMT1 and CMT2A [11], and some genetic forms of Parkinson’s and Alzheimer’s diseases [11]. Chimera C likewise improved mitochondrial motility in primary reprogrammed motor neurons from CMT2A patients and dorsal root ganglion neurons from CMT2A (MFN2 T105M mutant) mice [101]. A similar enhancement of mitochondrial transport was observed after Chimera C treatment of primary reprogrammed motor neurons from ALS patients and dorsal root ganglion neurons from ALS (SOD1 G93A mutant) mice [105]. 

The Chimera class of mitofusin activators has limited in vivo utility due to rapid metabolism by hepatic microsomal enzymes, manifested in vivo by nearly complete first-pass hepatic metabolism [100]. Thus, while the use of one compound to evaluate responses to mitofusin activation across different disease models avoided confounding effects that would have been introduced if multiple compounds were tried across different diseases, a new class of mitofusin activators was needed to evaluate whether and how making mitochondria “better” might mitigate in vivo disease symptoms. Accordingly, the original triazol/urea mitofusin activator chemical structure was re-engineered, eliminating the triazol group and modifying the urea [100,102]. These changes simplified and decreased the mass of mitofusin activators, while reducing the susceptibility of mitofusin activators to chemical modification by liver microsomal enzymes. The resulting second generation of mitofusin activators, collectively referred to as phenylhexanamides, prove sufficiently stable in vivo for proof-of-concept studies of mitofusin activation in preclinical animal models of neurodegenerative diseases. 

The first such study introduced a novel preclinical model of CMT2A mice expressing the human CMT2A mutation MFN2 T105M, specifically in motor neurons [101]. These mice develop neuromuscular dysfunction and neuroelectrophysiological abnormalities similar to human CMT2A starting at ~30 weeks of age, with progressive deterioration of neuromuscular function until 50–60 weeks old. MFN2 T105M mice were permitted to develop the flow-blown CMT2A phenotype at an age of 50 weeks, at which time they were randomized to treatment with the short acting phenylhexanamide mitofusin activator MiM111 or vehicle by daily intramuscular injection [101]. Compared to vehicle treatment, MiM111 reversed functional, neuroelectrophysiological, histological and ultrastructural abnormalities after 8 weeks of treatment. Phenotype normalization after mitofusin activation was associated with enhanced neuron regrowth in vitro and in vivo [101]. A follow-up study compared the pharmacokinetics and pharmacodynamics of orally administered MiM111 to a novel, longer acting oral mitofusin activator designated CPR1, but detected no difference in therapeutic efficacy between transient “burst” activation with MiM111 vs. sustained continuous mitofusin activation using CPR1 [104]. Together, these studies established the feasibility of using pharmacological mitofusin activation to interrupt or reverse neurodegeneration. What was not addressed by this work is the critical question of whether mitofusin activation could have similar therapeutic value in neurodegenerative conditions, such as ALS, that are not caused by a primary abnormality of MFNs. 

The first trial of mitofusin activation in preclinical ALS administered MiM111 to SOD1 G93A mice at the same dose and route of administration that had reversed murine CMT2A [101,105]. The results of this trial were unambiguously negative: daily intramuscular injection of MiM111 had no effect on ALS phenotype progression measured every 10 days from 2 months of age until mouse death. MiM111 treatment did not affect the deterioration in RotaRod latency (the time to falling off a rotating cylinder); in inverted grip time; in an overall score of neuromuscular function (combining the results of an elevated ledge test, hindlimb clasping when raised by the tail, gait abnormalities and severity of spinal curvature); of neuroelectrophysiological testing; or of survival to the pre-terminal paralytic state. 

MiM111 is a short or “burst” mitofusin activator with a serum half-life after intramuscular administration of only ~2 h and plasma levels that fall below the predicted therapeutic threshold within 12 h after administration [101]. It was considered that disease mitigation by mitofusin activation in conditions such as SOD1 mutant ALS, which are not caused by a primary abnormality of mitofusins, might require sustained and continuous treatment to interrupt the postulated vicious cycle of mitochondrial degeneration. Accordingly, another trial with an identical treatment window and outcome metrics was initiated using a similar next-generation phenylhexanamide mitofusin activator, CPR1. Compared to MiM111, CPR1 has a longer plasma half-life, improved nervous tissue bioavailability and an extended pharmacodynamic effect (measured as the improvement in mitochondrial motility through sciatic nerve neurons) [102]. Indeed, a single oral dose of CPR1 (60 mg/kg) normalized mitochondrial motility in ALS SOD1 G93A mouse neurons for 24 h, whereas the same oral dose of MiM111 lost efficacy to correct mitochondrial dysmotility after 9–12 h [102]. Moreover, CPR1 administered twice daily via the oral route to ALS SOD1 G93A mice from the time of initial phenotype onset (60 days) evoked a substantial delay in disease progression: the time courses over which RotaRod latency, inverted grip time, the combined neuromuscular function score and motor neuron electrical potential amplitude and conduction velocity deteriorated were delayed by 23–60%, depending upon individual metric. Remarkably, the tempo of phenotype progression for inverted grip time, combined neuromuscular score and nerve conduction time diverged throughout the 80-day CPR1 treatment period (mouse age 60–140 days), suggesting that the benefits of mitofusin activation for these endpoints were accretive. 

As expected, neuromuscular phenotype improvement in SOD1 G93A mice treated with the CPR1 mitofusin activator was associated with a reversal or improvement of the hallmark mitochondrial abnormalities. Compared to vehicle, CPR1-treated ALS mice exhibited less mitochondrial fragmentation and depolarization, greater mitochondrial residency within neuromuscular synapses and an increased number of neuronal synapses in distal hindlimb gastrocnemius muscles. The latter observation implied that motor neuron death or die-back was delayed by CPR1 treatment, which is consistent with markedly reduced apoptotic TUNEL labeling of lumbar spine ventral horns containing hindlimb motor neuron soma. Mechanistically, the in vivo benefits of mitofusin activation on SOD1 G93A ALS mice were recapitulated in vitro: CPR1 reduced ROS content and cell death in, and enhanced the regrowth of, cultured spinal motor neurons and dorsal root ganglion sensory neurons derived from the ALS mice. Finally, to address the possibility that CPR1 benefits were either unique to the SOD1 G93A mutation or somehow limited to mouse neurons, its effects on mitochondrial respiration were assayed on directly reprogrammed motor neurons derived from human ALS patients carrying the SOD1 L38V and I113T mutations. CPR1 treatment did not alter the characteristic ALS-associated decrease in mitochondrial respiratory complex enzymes, but it did improve basal and ATP-linked respiration. This suggests that part of the benefit of enhancing mitochondrial fusion in ALS is the improved functional coupling of oxygen consumption to ATP synthesis, which can also explain reduced ROS elaboration after CPR1 treatment.

Taken together, the results of the pioneering study by Wang et al. using mitofusin transgenesis [97] and the recent study by Dang et al. using a small molecule mitofusin activator [105] provide strong support for the notion that improving mitochondrial fusion and motility can measurably and favorably impact the course of neuromuscular degeneration in at least some genetic forms of ALS. Both preclinical trials employed the SOD1 G93A mouse model of ALS, but there were important differences in terms of study design that affect study interpretation. The transgenic Mfn2 overexpression approach introduced a persistent intrinsic protective measure targeted to neurons, whereas small molecule mitofusin activation was a system-wide therapeutic initiated only after signs of ALS-linked neuromuscular degeneration emerged. The small molecule treatment approach has more clinical relevance, but from a purely scientific perspective it is intriguing that life-long neuron-targeted MFN2 transgenesis did not produce qualitatively or quantitatively better outcomes than small molecule mitofusin activation initiated after disease onset. In other words, there is no evident preventative effect of enhancing mitofusins in ALS; the benefit is to delay progression. This provides additional corroboration for the idea that mitochondrial damage in SOD1 mutant ALS is a secondary event that can, nevertheless, be a major factor contributing to neuromuscular degeneration. 

## 4. Concluding Thoughts—DRP1 Inhibition or Mitofusin Activation in ALS?

The studies reviewed above demonstrate benefits in SOD1 G93A mice and a variety of cultured cells for inhibiting mitochondrial fission and augmenting mitochondrial fusion. Both approaches can correct mitochondrial fragmentation in ALS. However, it is arguable that fragmented mitochondria are largely a visually obvious epiphenomenon, i.e., a biomarker of disease that, itself, contributes little to progression. Accordingly, simply correcting mitochondrial fragmentation should not be the therapeutic objective. Rather, the goal of mitochondrial dynamics-targeted therapeutics should be to address the overall underlying mitochondrial pathology in ALS and similar chronic neurodegenerative syndromes, i.e., the failure to deliver healthy mitochondria to, and remove damaged cytotoxic mitochondria from, terminal nerve endings. It is this biological lesion that impairs neuronal signaling, repair and regeneration, and activates programmed death signaling pathways, ultimately evoking neuronal die-back and neurogenic myoatrophy. This concept encompasses the postulated feed-forward cycle of mitochondrial degeneration, in which respiratory uncoupling and impaired mitophagic quality control engender an expanding population of neuronal mitochondria with a diminished capacity to produce ATP and that elaborate toxic levels of ROS. In ALS, this damaged subpopulation may be more prevalent in the neuronal periphery due to impaired mitochondrial transport. In this context, it may be useful to speculate about the comparative value of suppressing mitochondrial fission vs. enhancing mitochondrial fusion when attempting to correct mitochondrial dysdynamism.

As detailed above, mitochondrial fission is essential to mitochondrial replication and repair through the mitophagic elimination of asymmetric depolarized daughters. Accordingly, the general inhibition of mitochondrial fission is poorly tolerated. However, the fission inhibitor peptide P110 may selectively suppress DRP1-Fis1 interactions that are increased in, and contribute to, disease, while sparing physiological mitochondrial fission. This selective inhibitory effect on pathological mitochondrial fission likely contributes to the benefits of P110 in murine ALS. On the other hand, mitochondrial fusion is necessary for long-term mitochondrial respiratory fitness; is a mechanism for damage repair; and mitofusins play contextual regulatory roles in mitophagy and mitochondrial transport. Thus, enhancing mitofusin expression and/or activity has been well tolerated by mitochondria, their host cells, and by all tissues examined. In essence, suppressing fission can interrupt disease-related mitochondrial pathology, whereas enhancing mitochondrial fusion can increase mitochondrial resilience to and recovery from damage. Conceptually, these modalities appear more complementary than overlapping, and it seems likely that each approach would fill in therapeutic gaps left by the other. Notwithstanding the scientific importance of evaluating the potential additive or synergistic effects of fusion augmentation and fission inhibition in neurodegeneration, a preclinical evaluation of P110 vs. CPR1 vs. both in the SOD1 G93A mouse may never be performed because of the tremendous amount of work involved, the expense of doing so, and because it would almost certainly be viewed by many journals as “non-mechanistic”, “incremental” or “confirmatory”. We view this situation as an unfortunate reality. As new approaches and reagents continue to be introduced, there will be a growing need to compare and combinatorially evaluate multiple approaches to correcting mitochondrial defects. The long-term goal should be to better define the role of mitochondrial dynamics-targeted therapeutics among the many promising approaches that might form an ensemble therapeutic approach to attack ALS and other neurodegenerative conditions manifesting abnormal mitochondria.

## Figures and Tables

**Figure 1 cells-12-01188-f001:**
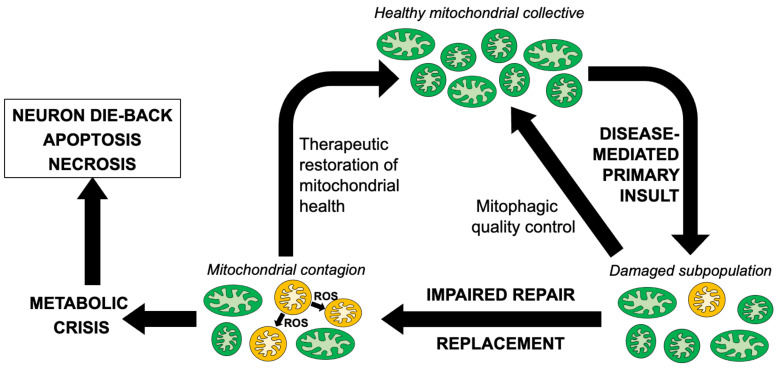
Schematic depiction of reparative mechansisms and the putative cycle of mitochondrial degeneration contributing to neuromuscular dysfunction in ALS and similar neurodegenerative conditions.

**Figure 2 cells-12-01188-f002:**
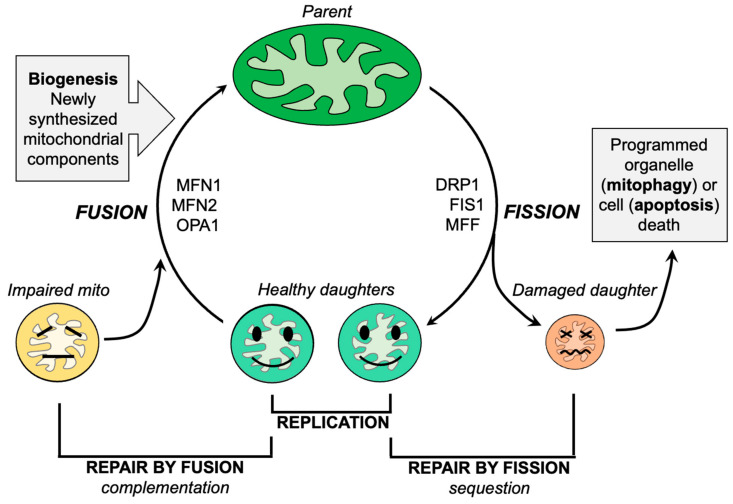
Schematic depiction of the roles of mitochondrial fission and fusion in maintaining mitochondrial homeostasis.

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
