# Peer review of "Reversing Dysdynamism to Interrupt Mitochondrial Degeneration in Amyotrophic Lateral Sclerosis"

_cells, 2023, doi:10.3390/cells12081188_

Round 1

Reviewer 1 Report

This manuscript reviews the many observations about the impaired mitochondrial transport and dynamics (fusion and fission), combinedly termed as “dysdynamism” in Amyotrophic Lateral Sclerosis (ALS) and how it contributes to the pathogenesis of the disease. Overall, the review is comprehensive and up to date. The later part of the review focuses on the therapeutic potential of correcting dysdynamism for ALS. While dysdynamism in neurovegetative diseases, including ALS, by itself has been reviewed on many occasions in detail, I find the therapeutic aspect part of the article interesting and a new addition to the literature. Thus, I feel that the manuscript is worth publication in Cells. Nevertheless, the manuscript will benefit from a careful proofreading.       

Reviewer 2 Report

The review by Dorn describes what is known about the role of mitochondria in ALS. This is a very important topic and a potential avenue for treatment. Therefore, this is a highly timely manuscript that will likely receive many citations. However, the text in its current form is not yet very helpful to uneducated readers. Many important concepts are not or only superficially introduced. The most significant shortcomings are detected in the context of ROS, the links between mitochondrial dynamics and movement and ER-mitochondria contacts. To become the resource this paper should be, as rightfully understood by the authors, these deficiencies should be thoroughly addressed. 

Specific Points:

1.    The function and significance of ROS seems central to this text. However, there is a very superficial description of how ROS are made within mitochondria and what they do. This is currently a serious deficiency. ROS production within CNS cell types is a big topic and multiple sources inside and outside mitochondria are important players in neurodegeneration (e.g., Nox2, Nox4). Also, regarding mitochondrial ROS production, several key studies by the Belousov lab have recently delineated their production, far from obligatory. Moreover, what are the biochemical consequences of ROS? Do they alter protein functions? Do they disrupt proteins or lipids? None of this is discussed but for such a key aspect of mitochondria in ALS, more is needed. 

2.    More mechanistic insight into mitochondrial movement is necessary, especially regarding the links between mitochondrial dynamics and movement. At the moment, it is not clear where the exact links are and how they could be pharmacologically targeted, when initially the text suggests that movement is particularly critical for the disease progression. Later in the last chapter, the authors suggests that mitochondrial dynamics is probably not a good therapeutic avenue. Therefore, the reader is left confused about what can and should be done in the context of improving mitochondrial functions. Which ones? Why?

3.    Line 153: More details are needed on how mitochondrial health was improved in the SOD1 model. Also, I the lines below therapeutic approaches concerning ROS and apoptosis control are mentioned but no details are provided. Why were these approaches not pursued? At what stage are they? Since the control of ROS and apoptosis impinges on mitochondrial dynamics this must be expanded. 

4.    Several mitochondrial dynamics proteins discussed in the text have important roles in ER-mitochondria contacts but this aspect is not discussed. It is puzzling that the role of calcium is not discussed. 

5.    There are multiple mechanistic links between ROS and mitochondrial dynamics (e.g., mitofusin-2, Drp1) but none are discussed. 

6.    Mechanistic insight into the triazol/urea activator of mitofusins is missing. Line 313-315: the link to cytochrome P450 is unclear. 

Minor points:

1.    Typos in line 87: anterograde and retrograde

Author Response

The review by Dorn describes what is known about the role of mitochondria in ALS. This is a very important topic and a potential avenue for treatment. Therefore, this is a highly timely manuscript that will likely receive many citations. However, the text in its current form is not yet very helpful to uneducated readers. Many important concepts are not or only superficially introduced. The most significant shortcomings are detected in the context of ROS, the links between mitochondrial dynamics and movement and ER-mitochondria contacts. To become the resource this paper should be, as rightfully understood by the authors, these deficiencies should be thoroughly addressed. 

Author’s response: Thank you. The challenge when I received the invitation to write this review was to come up with something conceptually new, given the abundance of scientific literature on ALS. My PubMed search identified 152 papers discussing the topic of ROS suppression/inhibition in ALS, and 155 papers dealing with the topic of apoptosis inhibition in ALS. By contrast, there are fewer than a dozen PubMed papers on the topic of correcting mitochondrial dynamic imbalance/dysdynamism in ALS.

My idea for providing a new perspective in this Cell review was to focus on ALS therapeutics, and specifically to take a new, but deep, look at the notion that correcting abnormal mitochondrial dynamism can mitigate the disease.  Joshi and Mochly-Rosen’s definitive review on this topic (in my opinion) was published just 5 years ago (reference 46 in the reviewed manuscript), and focused on the approach of suppressing mitochondrial fission. Thus, my aim here was to update the literature from the viewpoint of pharmacologically enhancing mitochondrial fusion, which was not known to be possible when the Joshi/Mochly-Rosen review was published.

Regarding the role of MFN2 (but not MFN1) in mito-ER contacts, I have nothing to add to what is already reported. To my understanding, MFN2 linkage of mito to ER or SR can contribute, but is not exclusively responsible for, privileged calcium microdomains that facilitate calcium cross-talk between organelles. How this phenomenon is perturbed in, or may contribute to, neuronal dysfunction/death in ALS in vivo is unclear. Our unblished results indicate that mitofusin activation neither disrupts nor facilitates corrects mito-ER coupling. Accordingly, the review focuses on those mechanistic aspects of mitofusin activation that we understand in ALS.

R2-1.    The function and significance of ROS seems central to this text. However, there is a very superficial description of how ROS are made within mitochondria and what they do. This is currently a serious deficiency. ROS production within CNS cell types is a big topic and multiple sources inside and outside mitochondria are important players in neurodegeneration (e.g., Nox2, Nox4). Also, regarding mitochondrial ROS production, several key studies by the Belousov lab have recently delineated their production, far from obligatory. Moreover, what are the biochemical consequences of ROS? Do they alter protein functions? Do they disrupt proteins or lipids? None of this is discussed but for such a key aspect of mitochondria in ALS, more is needed. 

Author’s response: As introduced above, there is an abundance of literature dealing with the possible role of ROS in ALS, and positing that ROS suppression can be therapeutic.  To re-hash what has been so thoroughly covered would, in my opinion, dilute the novel aspects of this review. In the revised manuscript I provide some additional context (pg 3) and refer interested readers to a recent comprehensive review.

R2-2.    More mechanistic insight into mitochondrial movement is necessary, especially regarding the links between mitochondrial dynamics and movement. At the moment, it is not clear where the exact links are and how they could be pharmacologically targeted, when initially the text suggests that movement is particularly critical for the disease progression. Later in the last chapter, the authors suggests that mitochondrial dynamics is probably not a good therapeutic avenue. Therefore, the reader is left confused about what can and should be done in the context of improving mitochondrial functions. Which ones? Why?

Author’s response: Clearly, the concluding paragraphs were confusing, in part because of some sentence fragments and ambiguities, which I’ve corrected. I did not mean to convey that mitochondrial dysdynamism is not a good therapeutic target. Indeed, a major point was that bringing mitochondrial fusion and fission into balance, and normalizing mitochondrial motility, is a therapeutic approach that can have ramifications far beyond simply correcting mitochondrial fragmentation. I’ve made extensive changes to the final section to clarify (pgs 10-11).

R2-3.    Line 153: More details are needed on how mitochondrial health was improved in the SOD1 model. Also, I the lines below therapeutic approaches concerning ROS and apoptosis control are mentioned but no details are provided. Why were these approaches not pursued? At what stage are they? Since the control of ROS and apoptosis impinges on mitochondrial dynamics this must be expanded. 

Author’s response: Additional detail and references have been provided on page 4. The mechanisms by which mitofusin activation improved mitochondrial health in the SOD1 mutant mice and patient reprogrammed neurons are detailed on pages 9 and 10.

R2-4.    Several mitochondrial dynamics proteins discussed in the text have important roles in ER-mitochondria contacts but this aspect is not discussed. It is puzzling that the role of calcium is not discussed. 

Author’s response: ER-mito calcium signaling is not the topic; mitochondrial fusion, fission and motility, which I call mitochondrial dynamics, is the topic. As has been well described and reviewed elsewhere, mitochondrial dynamics proteins tend to be multi-functional. For example, MFN2 participates in or regulates mitochondrial fusion, tethering to ER and lysosomes, mitophagy, apoptosis and mitochondrial transport. Here, I focused on mitochondrial dysdynamism because its correction using several different approaches (small molecule mitofusin activators, small molecule DRP1 inhibitor, peptide DRP1 inhibitor, forced MFN2 expression) has mitigated some aspects of ALS in preclinical models. The reviewer may be interested to know that our unpublished data indicate small molecule mitofusin activators do not alter mito-ER contacts.

R2-5.    There are multiple mechanistic links between ROS and mitochondrial dynamics (e.g., mitofusin-2, Drp1) but none are discussed. 

Author’s response: Again, ROS is a topic that has been extensively covered in the published literature, and I was trying to present a new/different perspective. As noted above, in the revision I’ve referred interested readers to excellent reviews on this topic.

R2-6.    Mechanistic insight into the triazol/urea activator of mitofusins is missing. Line 313-315: the link to cytochrome P450 is unclear. 

Author’s response: If I understand correctly the reviewer is asking for a more detailed explanation of how the initial class of small molecule mitofusin activators “work”, and of the evolution from the prototypes to the current generation of compounds that are pharmaceutically acceptable for in vivo use? I have expanded this on page 8.

            CytP450 enzymes detoxify and clear many circulating substances, including drugs. To avoid further confusion I’ve removed the reference to cytP450 and simply indicate that chemical simplification reduced modification by hepatic enzymes (pg 8).

Minor points:

  1. Typos in line 87: anterograde and retrograde

Author’s response: corrected, thanks.

Round 2

Reviewer 2 Report

I have read the revised version of this review and recognize the intent of the author. This takes care of all points previously raised, with one exception. Given ROS are key to ALS, and the authors wants to focus solely on mitochondrial dynamics, the following papers should be discussed in the text, perhaps in the introduction, since they make a direct link between the two topics: PMID: 29924999 and 29212658. 

Author Response

Cells ALS review R2

Reviewer 2

I have read the revised version of this review and recognize the intent of the author. This takes care of all points previously raised, with one exception. Given ROS are key to ALS, and the authors wants to focus solely on mitochondrial dynamics, the following papers should be discussed in the text, perhaps in the introduction, since they make a direct link between the two topics: PMID:

Author response

I now reference these two papers, which have nothing to do with ALS and one of which is fairly controversial, on pages 179,180 of the second revision.